# Simultaneous Genetic Ablation of PD-1, LAG-3, and TIM-3 in CD8 T Cells Delays Tumor Growth and Improves Survival Outcome

**DOI:** 10.3390/ijms23063207

**Published:** 2022-03-16

**Authors:** Elisa Ciraolo, Stefanie Althoff, Josefine Ruß, Stanislav Rosnev, Monique Butze, Miriam Pühl, Marco Frentsch, Lars Bullinger, Il-Kang Na

**Affiliations:** 1Experimental and Clinical Research Center, Max Delbrück Center for Molecular Medicine and Charité Universitätsmedizin Berlin, 13125 Berlin, Germany; elisa.ciraolo@charite.de (E.C.); stefanie.althoff@charite.de (S.A.); josefine.russ@bih-charite.de (J.R.); monique.butze@charite.de (M.B.); miriam.puehl@charite.de (M.P.); lars.bullinger@charite.de (L.B.); 2Department of Hematology, Oncology and Tumor Immunology, Charité-Universitätsmedizin Berlin, Corporate Member of Freie Universität Berlin, Humboldt-Universität zu Berlin, and Berlin Institute of Health, 10117 Berlin, Germany; stanislav.rosnev@charite.de (S.R.); marco.frentsch@bih-charite.de (M.F.); 3Berlin Institute of Health Center for Regenerative Therapies, Charité-Universitätsmedizin Berlin, 13353 Berlin, Germany; 4German Cancer Consortium (DKTK), 10117 Berlin, Germany

**Keywords:** adoptive immunotherapy, T lymphocytes, CRISPR/Cas9, checkpoint inhibitory molecule

## Abstract

Immune checkpoint inhibitors (ICI) represented a step forward in improving the outcome of patients with various refractory solid tumors and several therapeutic regimens incorporating ICI have already been approved for a variety of tumor entities. However, besides remarkable long-term responses, checkpoint inhibition can trigger severe immune-related adverse events in some patients. In order to improve safety of ICI as well as T cell therapy, we tested the feasibility of combining T cell-based immunotherapy with genetic disruption of checkpoint molecule expression. Therefore, we generated H-Y and ovalbumin antigen-specific CD8^+^ T cells with abolished PD-1, LAG-3, and TIM-3 expression through CRISPR/Cas9 technology. CD8^+^ T cells, subjected to PD-1, LAG-3, and TIM-3 genetic editing, showed a strong reduction in immune checkpoint molecule expression after in vitro activation, while no relevant reduction in responsiveness to in vitro stimulation was observed. At the same time, in B16-OVA tumor model, transferred genetically edited OT-1 CD8^+^ T cells promoted longer survival compared to control T cells and showed enhanced expansion without associated toxicity. Our study supports the notion that antigen-specific adoptive T cell therapy with concomitant genetic disruption of multiple checkpoint inhibitory receptors could represent an effective antitumor immunotherapy approach with improved tolerability profile.

## 1. Introduction

Immunotherapy including inhibitors of immune checkpoint molecules (also referred to as T cell inhibitory receptors, TIRs) has become an essential component in the management of solid tumors [1]. Since May 2006, multiple checkpoint inhibitors have been approved for various tumor entities and now represent the standard-of-care treatment for several cancer entities including melanoma, non-small cell lung cancer (NSCLC), urothelial carcinoma, renal cell carcinoma, and others [2,3]. Immunotherapy has led to a paradigm shift in cancer treatment, offering patients (with advanced and refractory cancer) the prospect of long-term survival and durable response. However, despite the initial enthusiasm regarding these durable responses in some patients, a significant clinical challenge to the successful application of immune checkpoint inhibitors (ICI) remains the moderate objective response rate of around 15–25% in most patient cohorts treated with mono-therapeutic regimens [4,5,6,7,8,9,10,11]. To address and overcome these challenges, ICI combination therapies with conventional chemotherapy and additional checkpoint inhibitors have already been approved and are currently intensively studied in preclinical as well as clinical settings [12,13]. Upregulation of other immune checkpoint molecules (on T cells) such as T-cell immunoglobulin and mucin domain-3 protein (TIM-3) and lymphocyte-activation gene 3 (LAG-3) has been reported in patients exhibiting secondary resistance to ICI [14,15] and it has been defined as a possible mechanism towards escape from the immune checkpoint blockade. TIM-3 has not only been found on tumor-infiltrating lymphocytes after PD-1 blockade, but its expression on cells of the innate immune system has also been reported [14,15]. Upon interaction with different ligands, TIM-3 is capable of inducing T cell tolerance, apoptosis of Th1 cells [16], or T cell exhaustion during chronic infection [17]. TIM-3 blockade has been shown to increase cytokine production and proliferation of tumor antigen-specific T cells [18] and combined blockade of TIM-3 and PD-1 has been demonstrated to improve anti-tumor responses in preclinical models and to revert tumor-induced immune suppression [19,20]. Another player in T cell regulation is LAG-3, a transmembrane receptor belonging to the Ig superfamily. LAG-3 is expressed on several subsets of tumor-infiltrating lymphocytes such as activated CD8^+^ T cells and Treg cells, as well as on myeloid-derived suppressor cells [21]. In addition, the expression of LAG-3 is often associated with other immune checkpoints, such as CTLA-4 and PD-1 [21]. Triggering of LAG-3 causes suppressed T cell activation and T cell-mediated cytokine secretion, which functionally synergize with the immunosuppressive activity of PD-1 to promote tumor immune escape [22,23]. Recent clinical studies evaluating the therapeutic potential of [23] LAG-3 blockade have revealed that the inhibition of LAG-3 as monotherapy might not be sufficiently effective in reverting the tumor-dependent immune suppression. These data suggest that LAG-3 therapy may be more potent when combined with other anti-cancer treatments [24,25,26,27,28]. On the other hand, several bodies of evidence support the use of TIM-3 antibodies in combination with anti-PD-1 therapy and multiple phase 2 clinical trials are currently recruiting (NCT04140500, NCT04080804, NCT04785820, https://clinicaltrials.gov/, accessed on 15 March 2022) [29]. 

Although these TIRs represent a promising target for tumor immunotherapy, they still have a fundamental role in regulating an appropriate immune response and avoiding autoimmune reactions. Therefore, even if the potent synergy between PD-1, LAG-3, and TIM-3 highlights a promising immunotherapeutic combination for future treatments, a thin borderline between efficacy and toxicity of TIRs inhibition exists. Everyday clinical practice showed us that immunotherapies result in unique profiles of adverse immune events that differ from the typical toxicities of other cancer therapies. These include immune-related adverse events (irAEs) particularly in the skin, thyroid, intestine, and liver [30], which is in line with knockout animal models for single TIRs exhibiting severe autoimmune diseases ranging from increased susceptibility to fatalities [31]. Together with single knockout, other studies with a combined genetic ablation of TIRs showed exacerbations of autoimmunity diseases in prone genetic backgrounds. Accordingly, LAG-3 deficiency/blockade in the absence of PD-1 induces lethal myocarditis in BALB/c mice [32] as well as lethal autoimmune conditions in other genetic backgrounds [22]. Double knockout mice such as Lag3^−/−^Pdcd1^−/−^ mice were shown to develop a lethal autoimmune condition that results in high morbidity with up to ~80% of mice being moribund by 10 weeks [22]. 

All these data showed that a concomitant blockade of several TIRs even with remarkable efficacy in multiple types of cancer is often accompanied by systemic toxicity, which requires specific management and could invalidate the benefits of cancer immunotherapies. Consequently, cellular immunotherapy strategies have been developed to more specifically target antigens expressed by the tumor cells such as transgenic T cells carrying a tumor-specific T cell receptor (TCR) or CAR, which have already been approved for the treatment of B cell lymphoma, lymphoblastic leukemia, and myeloma [33]. 

Numerous preclinical studies assessed the anti-cancer effects of chimeric antigen receptor (CAR) T cells in combination with concomitant genetic editing of checkpoint molecules [34,35,36]. CAR T cells with an edited PD-1 gene, mediated via CRISPR/Cas9, have demonstrated a stronger antitumor activity than unedited CAR T cells in orthotopic mouse models of chronic myeloid leukemia, hepatocellular carcinoma, and glioma [34,35,36]. The absence of PD-1 in CAR T cells increases tumor infiltration and persistence and ultimately enhances tumor clearance [34,35,36]. However, most of preclinical studies on CAR T cells have been so far focused on single blockade of inhibitory receptors, and not much is known about the therapeutic potential and toxicity of simultaneous blockade of PD-1, TIM-3, and LAG-3. As an example, a recent study showed that triple short hairpin RNA (shRNA) mediated silencing of PD-1, LAG-3, and TIM-3 in CAR-T cells improved their antitumor function [37]. 

In view of the further progress in this therapeutic landscape, we investigated the feasibility of triple gene editing of checkpoint molecules (PD-1, LAG-3, and TIM-3) in T cells for adoptive T cell therapy (ATT), as a means to improve anti-tumor effects and simultaneously avoid immune-mediated adverse effects as caused by checkpoint blockade. In the present study we tested the in vitro anti-tumor cytotoxicity of genetically edited T cells and assessed their effect on the clinical outcome (e.g., survival, therapy-induced toxicities) of tumor-bearing recipients. 

## 2. Results

### 2.1. CRISPR/Cas9 Gene Editing Induces Reduction in PD-1, LAG-3, and TIM-3 Expression

In order to induce permanent disruption of PD-1, LAG-3, and TIM-3 expression in CD8^+^ T cells, we took advantage of the CRISPR/Cas9 system [26]. The crRNA sequences, necessary for inducing Cas9-specific double strain brakes, were designed for their binding to a specific exon of each target gene. Selection of more suitable sequences was performed by considering the cutting frequency determination (CFD) score and the absence of off-targets. PD-1_crRNAs were selected to bind the first exon while LAG3-3_crRNA and TIM-3_crRNA the second and third exon, respectively (Table 1). To test and validate the gene editing efficiency of the designed crRNAs (Table 1), the murine T lymphoblast cell line EL4 was transfected by nucleofector with RNPs containing the Cas9 protein and the preassembled PD-1, LAG-3, and TIM-3 crRNA with the trRNA. After transfection, EL4 cells were expanded and single cell clones were isolated by using limiting dilution in a 96-well plate. In order to identify clones with CRISPR/Cas9-induced modifications of PD-1, LAG-3, and TIM-3, every clone was lysed and the expression of PD-1, LAG-3, and TIM-3 was analyzed by western blot (Appendix A). The expression level of each protein was calculated over the expression of GAPDH. All clones that showed a reduction in the protein higher than 30% were isolated, expanded, and subjected to DNA sequencing, in order to prove the presence of the induced genetic modification. DNA sequencing analysis revealed that all clones with less than 50% of PD-1, LAG-3, and TIM-3 expression presented a DNA modification at the site of action of the Cas9 enzyme (Appendix A), however, no specific mutations were identified.

The CRISPR/Cas9 system was further tested on CD8^+^ T cells derived from MataHari BLITC (MH) [38] and OT-1 transgenic (OT-1) [39] mouse models characterized by the expression of a TCR, specific for the H2-Db-restricted epitope of histocompatibilty Y (HYA, known as H-Y) and for ovalbumin peptide residues 257-264 (OVA257-264), respectively. CD8^+^ T cells were initially isolated from the spleen through CD8^+^ negative selection and, to induce in vitro expansion, activated in culture for 24 h with anti-CD3/CD28 antibodies, in the presence of IL-2. After one-week expansion in the presence of IL-7/IL-15 [40], CD8^+^ T cells were transfected by nucleofector with RNPs containing the Cas9 protein and the preassembled crRNAs with a labeled ATTO™550trRNA (Table 1). T cells were subsequently cultivated for a further three days in the presence of IL-7 and IL-15 for resting and subsequently subjected to long-term in vitro activation (72 h), in order to trigger the expression of TIRs.

We first analyzed whether transfection as well as tripled CRISPR-Cas9 editing could influence the ability of MH and OT-1 CD8^+^ T cells to expand in vitro. As shown in Appendix A, all three groups of CD8^+^ T cells expanded over a period of 14 days in a comparable manner. Subsequently, we stimulated MH CD8^+^ T cells with splenocytes derived from Rag^−/−^ immunocompromised mice that had been previously pulsed with H-Y peptide (UTY). After 72 h of incubation, PD-1, LAG-3, and TIM-3 membrane expression was assessed by flow cytometry analysis (Appendix A). The long-term in vitro stimulation of control MH CD8^+^ cells was able to induce increased membrane expression of LAG-3, while it failed to trigger the upregulation of PD-1 and TIM-3. In contrast to the control MH CD8^+^ T cells, the tripled CRISPR-edited MH CD8^+^ T cells (3KO) did not show any upregulation of LAG-3, thus demonstrating the effectiveness of the gene editing. To test also for PD-1 and TIM-3, we activated MH CD8^+^ T cells via anti-CD3/CD28 stimulation. As shown in Figure 1A, anti-CD3/CD28-mediated activation was able to induce the expression of PD-1, LAG-3, and, albeit to a lesser extent, TIM-3 (control MH CD8^+^ T cells). Together, genetically modified MH CD8^+^ T cells exhibited significantly diminished expression of TIRs upon peptide-specific activation (Appendix A) and non-specific activation (Figure 1A) compared to the controls (not transfected (N.T.) and transfection control (T.C.). 

Subsequently, we wanted to analyze the reproducibility of the phenotype induced by our CRISPR/Cas9 gene editing system, by applying it to further transgenic antigen-specific CD8^+^ T cells, namely to OVA-peptide stimulated OT-1 CD8^+^ T cells. After a 72 h long stimulation of OT-1 cells with OVA peptide pulsed Rag^−/−^ splenocytes expression of PD-1, LAG-3, and TIM-3 was suppressed in 3KO OT-1 CD8^+^ T cells compared with non-gene-edited OT-1 CD8^+^ T cell controls (N.T. and T.C.) (Figure 1B). Together, our data demonstrate the effectiveness of the CRISPR/Cas9 gene editing system in concomitantly inhibiting the expression of PD-1, LAG-3, and TIM-3 in different transgenic CD8^+^ T cells. 

### 2.2. Comparable Immune Function and Anti-Tumor Activity between Triple CRISPR/Cas9 Genetically Edited CD8^+^ T Cells and Control

In order to test the immune functionality of CD8^+^ T cells after gene editing, isolated MH CD8^+^ T cells were transfected with a CRISPR/Cas9 system and their ability to produce inflammatory cytokines, in particular IFNγ, was compared with not transfected control CD8^+^ T cells. Following the protocol described in the preceding results section, supernatant of MH CD8^+^ T cells with splenocytes pulsed with the UTY peptide were collected and the production of IFNγ was measured via ELISA. Our data showed that gene-edited 3KO MH CD8^+^ T cells and transfection-control T cells produced comparable levels of IFNγ to not transfected control T cells (Figure 2A). Furthermore, 72 h stimulation with anti-CD3/CD28 antibodies caused production of comparable levels of IFNγ in 3KO MH CD8^+^ T cells as well as in the corresponding control T cells (N.T. and T.C., Appendix A). In addition, a similar trend towards comparable responsiveness to immune stimulation was observed for the production of IFNγ upon ova-peptide stimulation in 3KO OT-1 CD8^+^ T cells as compared to control T cells. These data confirm that the general functionality of T cells regarding cytokine expression upon stimulation was not affected by the genetic editing of the TIRs (Figure 2B). 

In the next step, we determined the anti-tumor cytotoxic activity of CD8^+^ T cells after CRISPR/Cas9 genetic editing. Isolated MH CD8^+^ T cells were activated for 72 h with anti-CD3/CD28 antibodies and subsequently incubated with an H-Y expressing murine male bladder carcinoma cell line, MB49. Different rates between MH CD8^+^ T cells and MB49 tumor cells were analyzed and cytotoxicity was quantified through MTT cell viability assay at different time points. The percentage of surviving cells (% of MB49) was calculated in relation to MB49 alone (Figure 2C). In all three groups, MH CD8^+^ T cells showed similar levels of cytotoxic activity against MB49 with killing rates of more than 80% for the 10:1 rate. In addition, no differences were identified in the cytotoxic reaction between different groups, at different time points and among the different rates of CD8^+^/MB49. Comparable results were obtained with OT-1 CD8^+^ T cells, whose cytotoxicity was tested against B16 melanoma cells, expressing ovalbumin (B16-OVA). As for MH T cells, OT-1 CD8^+^ T cells with and without edited expression of PD-1, LAG-3, and TIM-3 showed comparable cytotoxicity against tumor cells (Figure 2D). In order to confirm the status of TIRs expression upon tumor cell-mediated antigen exposure during the killing assay, gene-edited 3KO MH and OT-1 CD8^+^ T cells, respectively, were isolated, stained for PD-1, LAG-3, and TIM-3 and subjected to analysis by flow cytometry. As shown in Appendix A, tumor-specific activation did not induce the expression of PD-1, LAG-3, and TIM-3.

### 2.3. Triple Editing of PD-1, LAG-3, and TIM-3 Improves the Anti-Tumor Activity of OT-1 CD8^+^ T Cells In Vivo

To determine the in vivo ability of 3KO CD8^+^ T cells to engraft, persist, and mediate anti-tumor activity, genetically edited (3KO), transfected control (T.C.), and not transfected (N.T.) CD8^+^ T cells were transferred into irradiated immunocompetent female recipients bearing subcutaneous B16-OVA tumors (5–9 mm^3^).

Tumor growth was measured every two days. Two different experimental endpoints were defined: (a) the tumor exceeded a volume of 1500 mm^3^ and (b) ex vivo tumor analysis, 16 days after ATT. Reduced as well as delayed tumor growth was observed in mice engrafted with 3KO OT-1 CD8^+^ T cells as compared to controls (Figure 3A). Furthermore, the overall survival of mice upon transfer of 3KO OT-1 CD8^+^ T cells was significantly higher, with 75% survival rate at day 16 (Figure 3B). In order to determine whether higher tumor infiltration by 3KO OT-1 CD8^+^ cells was correlated with the observed tumor growth retardation and increased survival, tumors were analyzed for OVA-peptide specific T cells (Vβ5, β-chain of the OVA-specific TCR) upon reaching a volume of 1500 mm^3^ or at day 16 after ATT. Flow cytometry analysis revealed a slightly but not significant increase in tumor-infiltrating 3KO Vβ5^+^ OT-1 CD8^+^ T cells compared to controls (Figure 3C). On the other hand, splenic 3KO Vβ5^+^ OT-1 CD8^+^ T cells were significantly elevated in comparison to control mice (Figure 3C). In addition, we tested in vivo the ability of triple edited OT-1 CD8^+^ T cells to infiltrate and persist in the tumor environment. Therefore, we took advantage of the OT-1 BLITC mouse model, where CD8^+^ T cells constitutively expressed Renilla Luciferase. Eight days after ATT, mice were anesthetized and injected with coelenterazine and the resulting luminescence was measured by using a Xenogen IVIS 200. As shown in Appendix A, the triple knockout triggered a stronger T-cell BLI signaling at peak in the tumor. 

Besides demonstrating the potential for improved anti-tumor activity of specific T cells through genetic editing of LAG-3, PD-1, and TIM-3, we also monitored the systemic toxicity induced by ATT with genetically edited T cells as compared to control T cells. To this aim, mice, bearing B16-OVA tumors and subjected to ATT with 3KO OT-1 CD8^+^ T cells were longitudinally evaluated via routine health and animal welfare monitoring for specific signs of systemic toxicity. In particular, the (subject-specific) weight was recorded and a clinical score including characteristics of hair, skin, and activity/vitality was assigned every 2–3 days. Ultimately, all animals maintained a good general health condition during the experiment with animals subjected to ATT with 3KO and control OT-1 CD8^+^ T cells demonstrating similar weight (Figure 3D) and clinical score dynamics (data not shown).

## 3. Discussion

Inhibition of TIRs is a promising effective immunotherapy, but success is obscured by therapy-induced irAEs. 

In order to address the possibility of reducing immunotherapy-associated toxicity while maintaining therapeutic efficacy, we investigated the feasibility of combining the genetic editing of TIRs, in particular PD-1, LAG-3, and TIM-3, as major players in the inhibition of T cell activity [41], to ATT. Since CRISPR/Cas9 technology has been already explored for multiplexing editing of a combinatorial set of genomic sites in T cells [42], we initially tested our CRISPR/Cas9 approach for simultaneous editing of PD-1, LAG-3, and TIM-3 on the T cell lymphoma cell line EL4 and then on primary T cells. Although sequence analysis of the identified clones with reduced TIR expression revealed expected genetic modifications in the area of the Cas9-specific cleavage site (PAM), an overlap of different genetic alterations was found, which suggested a genetic chimerism induced by the CRISPR/Cas9 system. Such chimerism has been already described in the literature [43,44,45], and it could be explained by the intrinsic activity of the repair machinery, that in the setting of double strand breaks acts differently on different alleles thus generating intra-allelic mosaicism [43,44,45]. Due to genetic chimerism and the difficulty of obtaining isolated single primary T cells and then exclusively expanding specific T cell clones, the efficacy of the multiple CRISPR/Cas9 editing strategy was tested by phenotypic analysis. In our system, CRISPR/Cas9 genetic editing was effective in significantly reducing the expression of PD-1, LAG-3, and TIM-3 on the plasma membrane of CD8^+^ T cells after prolonged immune activation induced by peptide-specific or CD3/CD28-mediated stimulation. We intentionally accepted not having a full edit of all T cells, as the application of a bulk approach would be more feasible to implement in clinical settings than to pursue triple-editing on single-cell level.

Regarding the in vitro functionality of genetically edited CD8^+^ T cells, we tested cytokine production after peptide-specific activation. Our aim was to demonstrate that genetic editing of PD-1, LAG-3, and TIM-3 is not harming functionality of CD8^+^ T cells. After in vitro activation with anti-CD3/CD28 antibody and antigen-specific stimulation we did not identify any decline in IFNγ production in either OT-1 or MH genetically edited CD8^+^ T cells. Observations in line with our data have been reported in the literature, e.g., Zhang et al. did not detect any alteration in IFNγ production in LAG-3-knockout CAR T cells compared to control CAR T cells [33].

Subsequently, given that the genetic editing of TIRs did not induce alterations in effector cytokine expression (upon in vitro stimulation), we studied the performance of the genetically edited T cells in vivo. Since OT-1 CD8^+^ T cells were more prone than MH CD8^+^ T cells to induce antigen-mediated TIRs expression, we tested anti-tumor activity of genetically modified OT-1 CD8^+^ T cells in B16 melanoma-bearing mice. Over a period of 16 days, mice subjected to ATT with 3KO OT-1 CD8^+^ T cells, showed slower tumor growth than control animals. At the same time, the overall survival rate significantly increased after simultaneous genetic editing of PD-1, LAG-3, and TIM-3, which is in line with previous literature showing reduced tumor growth and increased survival after ATT with double knockout (LAG-3 and PD-1) T cells [22,46]. The in vivo analysis revealed a persistence of tumor-specific (Vβ5) OT-1 CD8^+^ T cells with increased numbers of tumor infiltrating 3KO OT-1 CD8^+^ T cells as well as splenic lymphocytes. In concert, the increased Renilla Luciferase signal at the tumor site in the animals treated with 3KO OT-1 CD8^+^ T cells supports the conclusion that the genetic ablation of PD-1, LAG-3, and TIM-3 improves the ability of the CD8^+^ T cells to enrich in the tumor and to persist at the tumor site. Therefore, altogether, these data demonstrate that the simultaneous genetic ablation of PD-1, LAG-3, and TIM-3 expression induced a more sustained anti-tumor activity compared to non-edited T cells resulting in reduced tumor growth as well as increased survival. However, in comparison with the published knockout mouse model [22,46], in our study we detected a lower overall protection regarding tumor growth with no case of remission identified. This observation could be explained by the applied bulk approach comprising a pool of genetically edited OT-1 CD8^+^ T cells, possibly also including partially edited (e.g., single and double TIR-ablated T cells) as well as not-edited OT-1 CD8^+^ T cells. The co-transfection with ATTO™550 trRNA allowed the specific isolation of T cells positive for the crRNA/tracrRNA/Cas9 complex, but it did not give any indication on the percentage of successfully gene-edited T cells. Future studies should include supplementary measures that could be implemented in everyday clinical practice and which would further enable transfer of primarily gene-edited T cells for ATT, in order to increase the anti-tumor response. 

Likewise, clinically approved CD19 CAR T cell manufacturing also leads to the production of CAR T cell products with varied composition, which makes the measure of purity and identity (phenotypic signature, percentage of CAR T cells, and CAR expression) an essential part of the quality control process [21,47,48]. The transfer of T cells, with genetically ablated PD-1, LAG-3, and TIM-3 did not trigger any toxicity in our in vivo models which is in alignment with preclinical and clinical studies showing that a single KO for PD-1 in CAR T cells did not cause an increased incidence of serious adverse events [34,35,36]. Unfortunately, due to absence of a valid irAE animal model a head-to-head comparison of systemic toxicities triggered by systemic triple checkpoint blockade via antibody therapy (anti-PD-1, anti-LAG-3, and anti-TIM-3) on the one hand and intrinsically gene-edited T cells (3-KO for PD-1, LAG-3, and TIM-3) on the other was impossible [49,50,51]. 

Altogether, our study represents a first demonstration that the combination of T cell-mediated immunotherapy with genetic ablation of PD-1, LAG-3, and TIM-3 could delay tumor progression and significantly improve survival.

## 4. Material and Methods

### 4.1. CD8^+^ T Cell Isolation

Spleens of MH or OT-1 animals were harvested and dissected by mashing through a cell strainer (100 μm, Corning, New York, NY, USA, #10054-458) into a 50 mL tube under sterile conditions. After centrifugation (400× *g*, 5 min at RT) the cell pellet was resuspended in 1X Red Blood Cell (RBC, NH_4_Cl 155 mM, KHCO_3_ 10 mM, EDTA 0.1 mM). After 5 min the lysis was stopped by PBA (PBS with 0.5% BSA). Cells were counted via a hemocytometer, centrifuged (400× *g*, 5 min at 4 °C), and resuspended in MACS Buffer at a concentration of 2.5 × 10^9^ cell/mL. Subsequently CD8^+^ T cells were purified through CD8a^+^ T Cell Isolation Kit (Miltenyi Biotec, Bergisch Gladbach, North Rhine-Westphalia, Germany, #130-104-075). After purification, the cells were activated overnight by seeding them on a 6 well plate (up to 5 × 10^6^ per well), previously coated with murine anti-CD3 (3 μg/mL, clone 17A2, Invitrogen Waltham, MA, USA, #14-0032-82) and anti-CD28 (5 μg/mL clone E18 Biolegend, San Diego, CA, USA, #122002) antibodies. During activation, CD8^+^ T cells were incubated in TCM medium (RPMI, 10% FBS, 1 mM sodium pyruvate, 100 mM MEM nonessential amino acid, 5 mM HEPES, 100 U/mL penicillin, 100 mg/mL streptomycin) in the presence of interleukin-2 (IL-2, 1 ng/mL, Peprotech, Cranbury, NJ, USA, #200-02). CD8^+^ T cell were subsequently transferred into fresh 6 well plates and cultured in TCM with added interleukin-15 (IL-15, 50 ng/mL, Peptrotech, Cranbury, NJ, USA, #210-15) and interleukin-7 (IL-7, 10 ng/mL, Peprotech, Cranbury, NJ, USA, #217-17).

### 4.2. CRISPR/Cas9 and Nucleofection of EL4, MH and OT-1 CD8^+^ T Cells

Specific crRNAs (0.1 mmol/L each; PD1_crRNA1, TIM3_crRNA2, LAG3_crRNA3; Integrated DNA Technologies, IDT, Coralville, IA, USA) and universal transactivating crRNA (tracrRNA-ATTO™550, 0.1 mmol/L; Integrated DNA Technologies, IDT, Coralville, IA, USA) were mixed at equimolar concentrations and heated at 95 °C for 5 min in a ThermoMixer^®^ (Eppendorf, Hamburg, Hamburg, Germany). The mixture was then cooled down until room temperature (RT) was reached. Precomplexing of Cas9 (30 pmol/μL, TrueCut™ Cas9 Protein v2, Thermo Fisher Scientific, Waltham, MA, USA, #A36498) endonuclease with all crRNA/tracrRNA complexes (80 pmol/μL of PD1_crRNA1/tracrRNA, TIM3_crRNA2/tracrRNA, and LAG3_crRNA3 /tracrRNA) was completed by mixing and incubating for 10 min at room temperature. crRNA/tracrRNA/Cas9 mixture was electroporated into 2.5 × 10^5^ EL4 lymphoma cell line, 4 × 10^6^ MH and OT CD8^+^ T cells by 4D-Nucleofector^®^ X Unit (Lonza, Basel, Switzerland). As control of the transfection (T.C.) CD8^+^ T cells were transfected with a mock crRNA/trRNA/Cas9.

For MH and OT-1 CD8^+^ T cells nucleofection was performed 1 week after CD3/CD28 mediated activation. EL4, MH, and OT-1 CD8^+^ T cells positive for ATTO™550 were sorted and EL4 positive cells were cultured in RPMI 10% FBS, while MH and OT-1 CD8^+^ T cells in TCM with added IL-7 and IL-15.

### 4.3. Long CD3/CD28 and Peptide-Mediated CD8^+^ T Cell Activation

MH and OT-1 CD8^+^ T cells were stimulated with pulsed splenocytes. Splenocytes were isolated from Rag^−/−^ spleen, washed, and resuspended in free RPMI, in the presence of UTY peptide (WMHHNMDLI, 5 µg/mL) or OVA peptide (SIINFEKL, 5 µg/mL) for MH and OT-1 CD8^+^ T cells, respectively. CD8^+^ T cells and splenocytes, either pulsed or not pulsed (not stimulated N.S.), were plated in 24 wells with a ratio of 1 to 5, and incubated for 72 h. For CD3/CD28 activation, MH CD8^+^ T cells were plated on 24 well plates, pre-coated with murine anti-CD3 (1 μg/mL, clone 17A2, Invitrogen Waltham, MA, USA, #14-0032-82) and anti-CD28 (1 μg/mL clone E18, Biolegend, San Diego, CA, USA, #122002) antibodies. CD8^+^ T cells’ long-term activation, with peptide and CD3/CD28, was induced for 72 h in the presence of TCM with added interleukin 2 (IL-2, 1 ng/mL, Peprotech, Cranbury, NJ, USA, #200-02). After activation, MH and OT-1 CD8^+^ T cells were collected and stained with anti-CD3 (CD3e_PerCP-Cy5.5, Clone 145-2C11, Biolegend, San Diego, CA, USA), CD8 (CD8a_ APC-Cy7, clone 53-6.7, Biolegend, San Diego, CA, USA), LAG-3 (CD223_ BV421, clone C9B7W, Biolegend, San Diego, CA, USA), PD-1 (CD279_PE, clone HA2-7B1, Miltenyi Biotec, Bergisch Gladbach, North Rhine-Westphalia, Germany,), TIM-3 (CD366_ PE-Cy7, clone RMT3-23, eBiosciences, Thermo Fisher Scientific, Waltham, MA, USA) fluorescent antibodies and analyzed through flow cytometry (LSRFortessa™ Cell Analyzer, BD Biosciences, New York, NJ, USA).

### 4.4. Measurement of IFNγ Production

After long-term activation with the UTY and OVA peptide, supernatant was harvested and the production of IFNγ was measured through ELISA assay. Briefly, a 96 flat-bottom well plate was coated with an IFNγ capture antibody (1 µg/mL in PBS of anti-mouse IFNγ antibody, clone R4-6A2, BD Biosciences, New York, NJ, USA) and incubated overnight at 4 °C. After washing with PBS-T (PBS 1X, 0.1% Tween), wells were blocked for 30 min at room temperature (RT) with assay buffer (PBS 1X, 10% FBS). Standard solution was prepared by diluting albumin in order to have a concentration range from 4000 to 64 pg/mL. Subsequently, supernatant was applied to 96 wells in duplicate and incubated for 2 h at RT. Plates were then washed several times and wells were subsequently incubated with the detection antibody (1 µg/mL in PBS of Biotin Rat Anti-Mouse IFN-γ, clone XMG1.2, BD Biosciences). After washing, wells were incubated for 30 min at RT with HRP Streptavidin (1:1000, #554066, BD Biosciences, New York, NJ, USA). The amount of IFNγ was detected by colorimentric HRP enzymatic reaction through ABTS (2,2’-Azinobis [3-ethylbenzothiazoline-6-sulfonic acid]-diammonium salt, Sigma-Aldrich, St. Louis, MO, USA, #A3219) substrate. Then, 1% SDS was added to stop the enzymatic reaction and absorbance was measured at 410 nm with an ELISA reader.

### 4.5. Killing Assay

Killing assay was performed in a 96 well plate, where B16 male-derived urothelial carcinoma cell line, MB49 (RRID:CVCL_7076, kindly provide by the Prof. Thomas Blankenstein’s research group), and B16 derived murine melanoma cell line transfected with the ovalbumin protein (OVA), B16-OVA (kindly provided by Dr. R. Dutton’s research group) were plated. After 18 h, different rates of MH and OT-1 CD8^+^ cells were seeded on MB49 and B16-OVA tumor cells, respectively. CD8^+^ T cells and tumor cells were incubated from 24 to 72 h and subsequently CD8^+^ T cells were removed, stained with anti-CD3 (CD3e_PerCP-Cy5.5, Clone 145-2C11, Biolegend, San Diego, CA, USA), CD8 (CD8a_ APC-Cy7, clone 53-6.7, Biolegend, San Diego, CA, USA), LAG-3 (CD223_ BV421, clone C9B7W, Biolegend, San Diego, CA, USA), PD-1 (CD279_PE, clone HA2-7B1, Miltenyi Biotec, Bergisch Gladbach, North Rhine-Westphalia, Germany, ), TIM-3 (CD366_ PE-Cy7, clone RMT3-23, eBiosciences, Thermo Fisher Scientific, Waltham, MA, USA) fluorescent antibodies and analyzed by flow cytometry. On the other hand, wells were washed and tumor cells’ death ratios were measured through MTT assay (Roche, Merck, Darmstadt, Hesse, Germany, #11465007001). The percentage of dead cells was calculated through the following formula: (1-(ABS_s/ABS_c)) × 100; ABS, absorbance; s, sample; c, control. 

### 4.6. Mice and OVA Tumor Model

All mouse experiments were conducted in compliance with the institutional guidelines of the Max Delbrück Center for Molecular Medicine (Berlin, Germany) and approved by the Landesamt für Gensundheit und Soziales Berlin, Germany (G0307/15 and G0068/21).

MataHari (MH) are transgenic mice with RAG-2^−/−^ background and VJ segments encoding the TCRα (Vα115-Jα16) and TCRβ (Vβ8.3-Jβ1.1) [38]. OT-1 Rluc are transgenic mice with RAG-1^−/−^ background and VJ segments encoding TCRα (Vα2) and TCRβ (Vβ5) [39]. All mice were kept under specific pathogen-free conditions following institutional guidelines. The 10^5^ B16-OVA tumor cells, diluted 1:1 with Matrigel^®^ (BD Biosciences, New York, NJ, USA, #356234), were subcutaneously injected into the right flank. Tumors were left to grow until a volume between 50 and 100 mm^3^. When the tumor reached this range of volume just before adoptive T cell transfer (ATT), Albino B6 mice were subjected to sub-lethal irradiation (5 Gy, RS2000 Biological Irradiator, Rad Source Technologies Inc, Buford, GA, USA). ATT was performed by intravenous injection of 2 to 3 × 10^5^ OT-1 CD8^+^ T cells (CRIPR/Cas9 edited, transfection control, and not transfected). After ATT, mice were monitored every two to three days, weighed, and tumor growth (volume) was recorded. As experimental endpoints, we defined the tumor volume (up to 1500 mm^3^) and the experiment duration upon ATT time (up to 16 days). Tumor diameters (width, W and length, L) were measured via a caliper and tumor volume (V) was calculated with the formula V = (W^2^ × L)/2 [52].

## Figures and Tables

**Figure 1 ijms-23-03207-f001:**
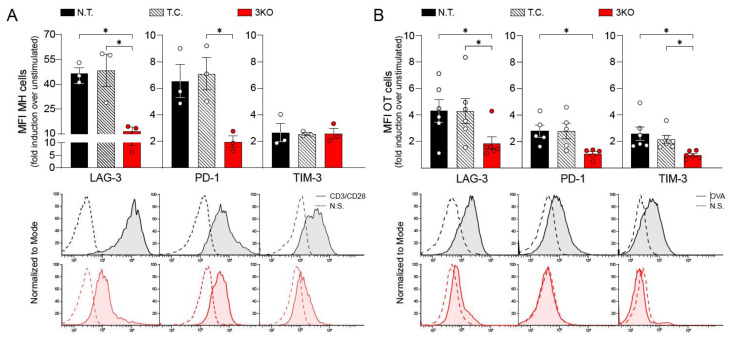
CRISPR/Cas9 gene editing and TIRs expression after 72 h CD8^+^ T cell stimulation. (**A**) Upper panel. Mean Fluorescence Intensity (MFI) of LAG-3, PD-1, and TIM-3 in MH CD8^+^ T cells after 72 h stimulation with anti-CD3/CD28 antibodies. Values correspond to the fold induction of CD3/CD28 stimulated MH CD8^+^ T cells over the unstimulated. Lower panel. Representative dot plot curves for PD-1, LAG-3, and TIM-3 intensity. Black line (straight and dotted) corresponds to N.T.; red line (straight and dotted) corresponds to 3KO. Bars are representative of three independent experiments with MH CD8^+^ T cells derived from three different animals. (**B**) Upper panel. Mean Fluorescence Intensity (MFI) of PD-1, LAG-3, and TIM-3 in OT-1 CD8^+^ T cells after 72 h stimulation with splenocytes incubated with OVA peptide. Values correspond to the fold induction of OVA stimulated OT-1 CD8^+^ T cells over the unstimulated. Lower panel. Representative dot plot curves for PD-1, LAG-3, and TIM-3 intensity. Black line (straight and dotted) corresponds to N.T.; red line (straight and dotted) corresponds to 3KO. Bars are representative of six independent experiments with OT-1 CD8^+^ T cells derived from six different animals. N.T., not transfected; T.C., transfection control; 3KO, Triple Knock Out for PD-1, LAG-3, and TIM-3 genes; N.S., not stimulated. Statistical analysis: one-way ANOVA with post-hoc test. * *p* < 0.05.

**Figure 2 ijms-23-03207-f002:**
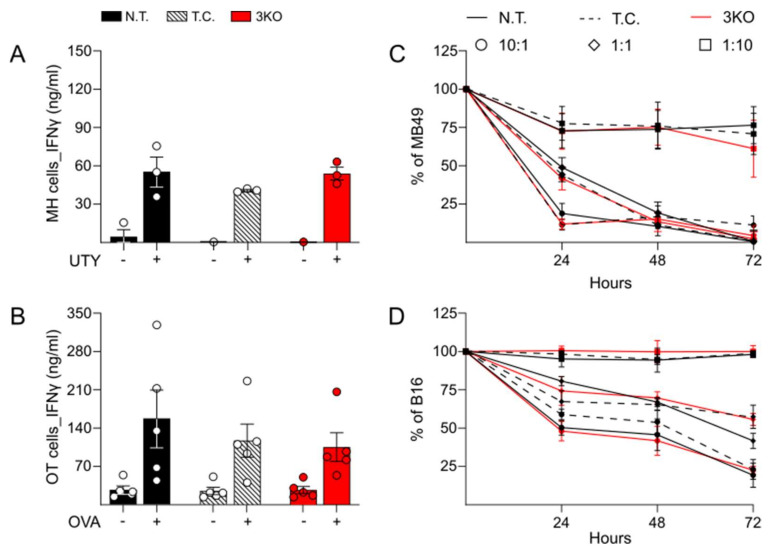
Immune activity and cytotoxicity in CRISPR/Cas9-edited MH and OT-1 CD8^+^ T cells. (**A**) ELISA quantification of IFNγ production after 72 h in presence or absence of UTY peptide. Bars are representative of three independent experiments with MH CD8^+^ T cells derived from three different animals. (**B**) ELISA quantification of IFNγ production induced by 72 h stimulation with OVA peptide in OT-1 CD8^+^ T cells. Data acquired from five different experiments conducted with OT-1 CD8^+^ T cells derived from five different animals. (**C**,**D**) MTT analysis of cytotoxic activity of MH and OT CD8^+^ T cells against MB49 and B16 tumor cells, respectively. Different rates between CD8^+^ T cells and tumor cells were analyzed: (10:1), 10^5^ CD8^+^ and 10^4^ tumor cells; (1:1), 10^4^ CD8^+^ and 10^4^ tumor cells; (1:10), 10^4^ CD8^+^ and 10^5^ tumor cells. Percentage of surviving tumor cells (% of MB49 and % of B16) was calculated on tumor cells not incubated with CD8^+^ T cells. Graphs are representative of CD8^+^ T cells derived from 3 different animals for each group. N.T., not transfected; T.C., transfection control; 3KO, triple knock out for PD-1, LAG-3, and TIM-3 genes.

**Figure 3 ijms-23-03207-f003:**
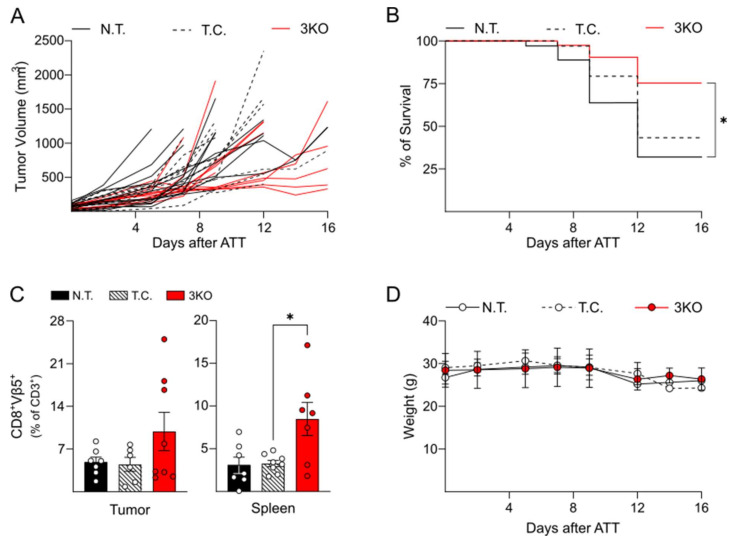
Effect of 3KO OT-1 CD8^+^ adoptive T cell transfer (ATT) on B16 tumor growth and survival. (**A**) Tumor growth after ATT of not transfected, transfection control, and triple CRISPR/Cas9 edited T cells. After ATT B16 tumors were measured every 2–3 days over a period of 16 days and mice were sacrificed after 16 days or when a tumor volume of 1500 mm^3^ was reached (experimental endpoints). (**B**) Survival rate after B16 xenograft. Percent of survival was calculated over a period of 16 days and statistical analysis was performed on a number of animals >6 for each group. (**C**) When one predefined experimental endpoint was reached, spleen and tumor were isolated, smashed through a cell strainer, and stained for CD3, CD8, Vβ5, PD-1, LAG-3, and TIM-3. Bars represent the percentage of CD8^+^Vβ5^+^ over total CD3^+^ T cells. Bars are representative of ≥6 animals per group. (**D**) Weight measurement after ATT every 2–3 days over a period of 16 days. N.T, not transfected; T.C. transfection control; 3KO, triple CRIPR/Cas9 edited OT-1 CD8^+^ T cells. Statistical analysis: one-way ANOVA with post-hoc test. * *p* < 0.05.

**Table 1 ijms-23-03207-t001:** List of crRNA used for CRISPR/Cas9 gene editing.

Gene	cRNA Name	RNA Sequence	PAM	Binding	CDF	Off-Targets
Pdcd1	PD1_crRNA1	ACAGCCCAAGTGAATGACCA	GGG	Exon 1	87	0-0-3-18-140
Havcr2	TIM3_crRNA2	ATGTGACTCTGGATGACCAT	GGG	Exon 2	80	0-0-1-14-133
Lag-3	LAG3_crRNA3	ACCCGCACCCGGTCGCTACA	CGG	Exon 3	98	0-0-0-1-16
-	trRNA	AGCAUAGCAAGUUAAAAUAAGGCUAGUCCGUUAUCAACUUGAAAAAGUGGCACCGAGUCGGUGCUUU	-	-	-	-

PAM, protospacer adjacent motif, DNA region targeted for cleavage by the CRISPR system. CFD, cutting frequency determination score. It summarizes all off-targets into a number from 0 to 100, where a higher number correspond to fewer off-targets, and vice versa. Off-targetsnumbers represent the number of off-targets with 0, 1, 2, 3, 4, mismatches, respectively.

## Data Availability

Not applicable.

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
