# Peer review of "Simultaneous Genetic Ablation of PD-1, LAG-3, and TIM-3 in CD8 T Cells Delays Tumor Growth and Improves Survival Outcome"

_ijms, 2022, doi:10.3390/ijms23063207_

Round 1

Reviewer 1 Report

The article “Simultaneous Genetic Ablation of PD-1, LAG-3, and TIM-3 in CD8 T Cells Delays Tumor Growth and Improves Survival Outcome” explores the idea to improve adoptive T cell transfer by simultaneously knocking out the inhibitory receptors PD-1, LAG-3 and TIM-3. In general, this is a very interesting research subject since this approach would combine adoptive T cell transfer and checkpoint inhibition while avoiding the side effects of systemic checkpoint inhibition treatment. The manuscript is well written and easy to follow. 

In a first step, the authors show that while the triple knockout cells are composed of a mosaic of all combinations of the desired knockouts, they still express significantly less of all three inhibitory receptors upon stimulation. In a next step they show that these triple knockouts are still able to produce IFN-γ and exert cytotoxicity. In a last experiment they show that the transfer of these cells significantly improves survival compared to not transfected control T cells.

While the general idea and principle of this paper is very interesting, in my opinion there are some scientific flaws:

  1. As also cited in this manuscript, several papers already show that CAR T cells with a knocked-out PD-1 gene display stronger anti-tumor activity than unedited CAR T cells. Since PD-1 single knockout T cells are not used as a control in this manuscript, the novelty of this paper is reduced to the fact that it is possible to generate living triple knockout T cells and that injection does not cause any toxic side effects. The lack of this important control makes it impossible to conclude whether knocking out all  three inhibitory receptors offers any additional advantage to single knockouts.
  2. In CAR T cell transfer the capability of these cells to proliferate and expand is of high importance. A proliferation analysis of the triple knockouts is missing in this manuscript.
  3. When T cell cytotoxicity is analyzed in vitro the ratio of T cells to target cells is very important because at some point a saturation might be reached. Therefore, to see actual differences in cytotoxicity between two T cell species it could be better to analyze a gradient of different ratios and draw conclusions from the resulting plot in its linear phase of cytotoxicity increase. In this manuscript only a ratio of 10:1 is tested which in our experience is a very large amount of T cells, likely enough for less cytotoxic T cells to exert the same cytotoxicity as actual more cytotoxic ones.
  4. Several papers show that the B16-OVA melanoma barely responds to checkpoint inhibition. Even though some publications also show that radiation makes the tumors more sensible towards checkpoint inhibition, it would be helpful to have a more thorough characterization of the triple knockout cells to be able to differentiate whether the improved tumor killing is caused by the lack of inhibitory receptor binding in the TME or by a generally improved fitness of these cells.

Beside these major points it would also be good to see how long this effect keeps the tumor in check. The mice in the survival experiment were killed on day 16 where 75% of mice recieving triple knockout cells were still alive. It would also be helpful to get a proper quantification of the tumor volumes per group.

In summary I would recommend a major revision before accepting this manuscript. The authors should add/adjust the following points:

  1. More functional assays (proper cytotoxicity assay, proliferation assay, activation markers). These assays could also be done in vivo as well.
  2. A proper quantification of the tumor volume
  3. In the best case they would repeat the experiments with PD-1 single knockout cells as a control. This would greatly improve the value of this paper.

With kind regards,

Author Response

We thank reviewer 1 for the valuable comments and criticisms. With the attached letter, we provide our point-by-point answers to the reviewer's comments and the details of the revisions in the manuscript. All changes in the manuscript and supplementary materials are highlighted in yellow.

Reviewer 2 Report

The manuscript of Ciraolo and co-authors is devoted to implementation of one of the approaches to creating engineered T-cells that are exhaustion-resistant and thus could be exploited for T cell-based therapy. Authors took advantage of CRISPR/cas-technology to concomitantly disrupt 3 immune checkpoints (ICs/TIRs), in order to prevent potential activation of compensatory mechanisms after mono-inhibition. Although the authors submitted their paper as ‘Communication’ (which is explained by small sample size and the use of basic in vitro tests and basic evaluation of in vivo response in mouse model), but methodologically, it presents a completed piece of work (taking in account a variety of transgenic lines of cells/mice used in experiments) and can be considered as a completed exploratory stage of a more comprehensive research (e.g. with engagement of clinical material). In their work, Ciraolo et al verified the effectiveness of triple PD1+LAG3+TIM3 genetic editing, preservation of in vitro functionality of modified CD8 T (according to their ability to secrete IFNg and mediate cytotoxicity upon Ag-stimulation and anti-CD3/CD28 activation), and evaluated the overall in vivo effect of such cells transferred to mouse tumor models on general parameters of tumor growth and tolerability.

I’d like to note good argumentation researchers used to support their conclusions and their choice of experimental design (e.g. choice of bulk approach instead of single-cell level, or limitations their ‘in vivo part’ of work) in ‘Discussion’, thereby anticipating possible questions. The text is well-written and easily readable.

I can make the following suggestions (rather technical that essential):

  1. If the requirements to the text length allow, it seems better to include Suppl Fig 4 in Fig3C in the main body (despite the values didn’t reach significance) due to high importance of T-cell infiltrate. Along with this, I’d like to clarify the sentence in the caption to Suppl Fig2 (which is corresponded to UTY stimulation of MH cells) ‘Values correspond to the fold induction CD8+ of CD3/CD28 stimulated MH T cells over the unstimulated’ - this phrase is equal to that in Fig 1A caption, which indeed describes CD3/CD28 stimulation - is this a typo in Suppl Fig2 caption?

Line 228: reference to Fig2 goes without 2C and 2D.

  1. In the caption to Suppl Fig 1, it would probably be better to include some key moments about blotting procedure and sequencing (despite their routineness: antibodies, sequencing platform, etc.), analogously as the authors wrote in Suppl Fig 3 caption.
  2. In ‘Methods’ (‘CRISPR/Cas9 and nucleofection of…’), could the authors add a phrase defining Transfection Control (T.C.) - what type of control was used.
  3. Concerning the essence of the work, the described results and observations make reasoning about the impact of cell type/tissue type specificity. In first, specificity in relation to the type of tumor cells: researchers used bladder cancer and melanoma cell lines for in vitro cytotoxicity testing and in vivo transplantation, i.e. tumors which are generally accepted as hot and immune-responsive tumors (with FDA-approved ICI immunotherapy), but may one expect comparable effect in relation to tumor cells of another tissue origin?

Second, specificity in relation to the source/origin of CD8 T and mode of activation: as indicated in the text, «Since OT-1 CD8+ T cells were more prone than MH CD8+ T cells to induce antigen-mediated TIRs expression…..» (line 309), so, whether it means that there are some specific properties of genetic background that determine primary differences in TIRs’ expression pattern depending on the presence/absence of an antigen or co-stimulators. Surely, these questions rather address the huge problem of reliable predictors of physiological response to any type of ICs blockade, and can’t be solved within a single research.

In view of this, I’d also like to clarify the sentences « In order to confirm the status of TIRs expression upon tumor cell-mediated antigen exposure during the killing assay……..As shown in Supplementary Figure 3, tumor-specific activation did not induce the expression of PD-1, LAG-3, and TIM-3» (lines 231-232). Could the authors comment on it? (as it goes from the text, CD3/CD28 and Ag-specific stimulation of CD8T with pulsed splenocytes/APCs resulted in TIRs up-regulation in T.C. and comparable IFNg secretion both in T.C. and 3KO, while tumor cell-mediated Ag-stimulation resulted in similar activation of cytotoxicity both in T.C. and 3KO, but had no influence on TIRs’ status in all groups). Is there such a system in which such mode of CD8 T stimulation (i.e. tumor cell-mediated) could indeed result in induction of TIRs expression in control cells while its abrogation wouldn’t affect cytotoxic activity of 3KOs?

  1. In authors’ opinion, whether the results they obtained allow to evaluate objectively the contribution of TIM3 inactivation, if a noticeable change was seen only for OT-1 cells, while in case of MH CD8T, the extent of TIM3 expression changes was relatively small (as compared with PD1 and LAG3 ) both upon induction and after genetic ablation?

Author Response

We thank reviewer 2 for the valuable comments and criticisms. With the attached letter, we provide our point-by-point answers to reviewer's comments and the details of the revisions in the manuscript. All changes in the manuscript and supplementary materials are highlighted in yellow.

Round 2

Reviewer 1 Report

The authors have addressed my concerns. I have no further questions and recommend acceptance.